# The Effect of Blade Angle Deviation on Mixed Inflow Turbine Performances

**Mohammed Amine Chelabi** [1], **Milan Saga** [2], **Ivan Kuric** [3], **Yevheniia Basova** [4], **Sergey Dobrotvorskiy** [4],
**Vitalii Ivanov** [5,*] **and Ivan Pavlenko** [6]

1   FERTIAL SPA Company, Industrial Zone SPA BP 40, Arzew 31200, Algeria; chelabilma.usto@yahoo.com
2   Department of Applied Mechanics, University of Zilina, 8215/1, Univerzitna Str., 010 26 Zilina, Slovakia;
    milan.saga@fstroj.uniza.sk
3   Department of Automation and Production Systems, University of Zilina, 8215/1, Univerzitna Str.,
    010 26 Zilina, Slovakia; ivan.kuric@fstroj.uniza.sk
4   Department of Mechanical Engineering Technology and Metal-Cutting Machines, National Technical
    University "Kharkiv Polytechnic Institute", 2, Kyrpychova Str., 61002 Kharkiv, Ukraine;
    yevheniia.basova@khpi.edu.ua (Y.B.); sergiy.dobrotvorskyy@khpi.edu.ua (S.D.)
5   Department of Manufacturing Engineering, Machines and Tools, Sumy State University, 2,
    Rymskogo-Korsakova Str., 40007 Sumy, Ukraine
6   Department of Computational Mechanics Named after Volodymyr Martsynkovskyy, Sumy State University, 2,
    Rymskogo-Korsakova Str., 40007 Sumy, Ukraine; i.pavlenko@omdm.sumdu.edu.ua
*   Correspondence: ivanov@tmvi.sumdu.edu.ua

**Featured Application: This study contributes to the design of mixed inflow turbines to improve their performance in internal combustion engines.**

**Abstract:** The choice of blades for mixed turbines is to achieve the required deflection with minimal losses. In addition, it is necessary that the blade functions without a detachment in a wide area outside the nominal operating point of the machine. In the blade profile study, it is required to satisfy the conditions relating to fluid mechanics and those relating to the possibility of realization of construction. The work carried out presents the effect of the blade deviation angle on the geometric blade shape and the performance of the mixed inflow turbine on keeping the same rotor casing in order to improve its performances. It was remarked that the efficiency is proportional to the deviation angle's increase, but the rotor became heavy. It has been determined that the effect of the blade deviation angle on mixed inflow performances decreases dramatically starting from the angle $-20°$ for a 100% of machine load. It was urged to avoid relying on angles greater than $-20$ as values for blade deviation angles. The study noted that the maximum obtained in the output work and power is related to the highest the efficiency for a specific optimum design case ($-35°$ of deviation blade angle) due to the increase in the contact surface between the blade and the fluid, but the problem is that the rotor gets a little heavy (4.37% weight gain). Among recommendations, attention was given to the more significant absolute exit kinetic energies, for values of deviation blade angle between $-10°$ and $-20°$, where an exhaust diffuser is recommended to use to recover a part of it into a greater expansion ratio. These simulation results were obtained using a CFD calculation code-named CFX.15. This code allowed for the resolution of the averaged dynamic equations governing the stationary, compressible, and viscous internal flow.

**Keywords:** energy efficiency; industrial growth; mixed turbine; blade; hub; shroud; deviation

## 1. Introduction

The issues of intensifying technological processes and increasing the efficiency of mixed-flow turbines, which are also used in internal combustion engines, are a priority in modern mechanical engineering. The peculiarity of such turbines lies in the simultaneous presence of both axial and radial flow, and neither of these flows is negligible.

The mixed inflow turbine mainly consists of a volute, a rotor, and a diffuser. The volute is used to convert the potential energy into kinetic energy. It can be used as fixed blades at the outlet volute to direct the fluid at a specific angle. It is an established fact that the optimum absolute flow angle at the outlet volute is approximately $-13°$. The role of the rotor is to convert kinetic energy into mechanical energy, which is the subject of the proposed research. As for the diffuser, it converts the remaining kinetic energy into the potential energy, and it can be dispensed if there is an ideal energy transition in the rotor.

Due to the importance of the mixed flow turbine's role on the turbocharger used in internal combustion engines, many scientists are studying it. As a result of a deep analysis of works in this area, we have identified a number of interesting studies. Rajoo et al. [1] offered a comprehensive review of mixed-flow turbines. Watson et al. [2] compared the efficiency of axial, radial, and mixed turbines by different speed ratios, improving the turbocharged engine's performance with variable geometry. The potential improvements that the mixed flow turbine offers have been a great motivation for many research works on the application of ICE mixed flow turbines, such as Yamaguchi et al. [3], Chou et al. [4], Naguib [5], and Minegashi et al. [6]. In research [7], the authors argue that mixed flow turbines are considered the best option that offer significant benefits for automotive turbocharger applications. Wallace [8] symmetrically approached radial and mixed turbine design. Zhang et al. [9] tested mixed inflow turbine performances and compared them with the experimental results.

In parallel, the works of scientists who work in the field of studying the volute shape of mixed inflow turbine and its working conditions were considered. Wallace et al. [10,11] mentioned that the VGT (variable geometry turbine) is the next step in turbo compression technology at this time, and they conducted a series of VGT-equipped engine tests and examined the improvement achieved. They found that an "ideal" nozzle, specially designed for mixed-flow rotor entry, is essential for obtaining the maximum VGT performance. Baets et al. [12] used a pivoting nozzle vane to control the inlet mixed flow turbine. The recent development of VGT was experimentally tested by Rajoo et al. [13], which was designed to match the 3D of the mixed-flow turbine geometry at the leading edge. The design and the experimental data obtained by Rajoo [14] are used for validation throughout the current research. Gao Y. et al. [15] investigated stator end wall designs numerically for a mixed flow turbine. Lee et al. [16] designed and analyzed the volute of a radial and a mixed flow turbine. Ketata et al. [17] numerically studied a mixed flow turbine volute operating in various steady flow conditions. Maghnin et al. [18] presented the influence of volute cross-section shape on mixed inflow turbine performances. Lee et al. [19] analyzed the impact of a volute aspect ratio on the performance of a mixed flow turbine performance. Hamel et al. [20] investigated a twin entry flow turbine volute and benefits concerning the eco-system. Leonard et al. [21] studied the inlet mixed inflow turbine geometry. Ketata and Driss [22] studied the volute to rotor interspace loss on mixed turbine performance. Some researchers were interested in the operating conditions of mixed inflow turbines: Pesiridis [23] and Pesiridis and Martinez-Botas [24–26] modified the mechanism of actuation of sliding vanes to adapt to the nature of the pulsating flow of exhaust gas by inducing artificially sinusoidal motion. Wallace and Blair [27] reported the earliest experimental study of the irregular flow effect on mixed inflow turbine performance. Yamaguchi et al. [3] analyzed four rotors of inflow mixed turbines with different camber curves. Morrison et al. [28] studied the effects of flow conditions at rotor inlet on mixed flow turbine performance for automotive applications. In work [29], the unsteady rotor flow-field timescale in a highly loaded mixed flow turbine using validated 3-D unsteady computational fluid dynamics (CFD) carried out by ANSYS-CFX was quantitatively investigated. In paper [30], to control the shock wave and to improve the flow conditions of highly-loaded turbine cascades, an inverse method study on the S1 stream surface was conducted to weaken shock wave strength using the blade profile at the mid-span of the second stage rotor blade of an E3 turbine. Chen and Baines [31] concluded that the mixed inflow turbine design could be optimized by adjusting the volute outlet vortex angle to zero or a certain positive value. Padzillah

et al. [32] numerically and experimentally studied the flow angle effect on mixed inflow turbine performance and discovered that only one rotor blade part operates in optimal conditions. Karamanis et al. [33] studied the constant blade angle effect on the mixed inflow turbine performance in the steady and in the unsteady state.

Given the fact that the rotor shape has great importance, as it is the main element for the transformation of kinetic energy into mechanical energy, much attention has also been paid to this issue in modern literature. Palfreyman et al. [34] obtained the rotor geometry of the mixed flow turbine by radially scanning the leading edge of a radial turbine. This modification resulted in a blade curvature reduction of the secondary flow concerning its radial counterpart. Whitfield et al. [35] established a relationship between the blade angle, the cone angle, and the camber angle. Litim et al. [36] studied the effect of blade number on mixed inflow turbine performances. Leonard [37] presented mixed flow turbine rotor design and performance analysis with extended blade chords. Ketata et al. [38] tested seven designs of inflow mixed turbines with different numbers of blades from 3 up to 15 that have been considered for investigation and obtained porch results at Blair's work. Wallace et al. [39] based on one-dimensional analysis and empirical energy loss models to determine the optimal mixed inflow turbine rotor dimensions. Ke et al. [40] established a specific procedure for mixed flow turbine rotor design. Uswah et al. [41] optimized the hub and shroud by changing the position coefficient introduced into the Bezier polynomial to obtain better performance. Whitfield and Baines [35] developed a relationship (1) between cone angle, inlet blade angle, and camber angle.

$$\tan(\beta_{2b}) = \cos(\lambda) \ \tan g(\phi) \tag{1}$$

Chelabi et al. [42] studied the effect of cone and inlet blade angles on mixed inflow turbine performance in two cases, the first for a fixed outlet section volute and the second for a variable outlet section volute which was parallel to the rotor leading edge, some geometric parameters were fixed to keep the same rotor casing. Authors [43] analyzed the three-dimensional accelerating flow in a mixed turbine rotor. The studies [44,45] include research to ensure high-quality mechanical machining of turbine blades for regular operation.

Thus, the analysis of the current state of the issue shows a high interest in the research and manufacture of mixed-flow turbines due to their great promise. However, the complexity of the ongoing processes does not yet allow one to unambiguously determine the physical, technological and design parameters of products.

In particular, the relationship between the deviation blade angle and the shape of the rotor, and the performance of mixed-flow turbines has not been sufficiently studied. However, the importance of this parameter has been emphasized in many works.

Therefore, the search for optimal and rational parameters of turbines must be carried out at the design stage since the manufacture of turbines is an expensive process. This task can be accomplished by modeling the processes and structures of the developed turbines on digital models. Therefore, the present work aims to create a digital model of a mixed flow turbine and to analyze the effect of the deviation blade angle on the rotor shape and on the performance of a mixed flow turbine at various rotation speeds, which provide 50%, 75% and 100% performance. The main goal of the work is to increase the turbine's efficiency by increasing power and reducing the weight of its rotor. The body parameters remain unchanged.

## 2. Materials and Methods

### 2.1. Initial Rotor Design

Two existing techniques in progress are available to sufficiently meet our ambitions, which engage in the description of the blade geometry. Its complex configuration characterizes the latter at the level of the multiple curvatures in three-dimensional form arranged between the shroud and the hub. The first technique considers the projection of the structure in the perspective of the blade on parallel and perpendicular planes to the wheel's axis of rotation. The geometry is defined at the hub of a functional representation of the

two-dimensional curves. The second technique directly represents the geometry of the surfaces considered using a three-dimensional functional model by applying the Bezier polynomials mentioned in this article; this technique offers great interest for our choice to specify the blade geometry, which effectively adapts mixed inflow turbine. The meridian plane of rotor type A with constant blade angle [30]; Ref. [43] used in the mixed inflow turbines shown in Figure 1 is under examination [42]. The Bezier polynomial is used for this purpose. The relations below are respected (Equations (2) and (3)).

$$r = (1-u)^4 r_0 + 4u(1-u)^3 r_1 + 6u^2(1-u)^2 r_c + 4u^3(1-u)r_2 + u^4 r_3 \tag{2}$$

$$x = (1-u)^4 x_0 + 4u(1-u)^3 x_1 + 6u^2(1-u)^2 x_c + 4u^3(1-u)x_2 + u^4 x_3 \tag{3}$$

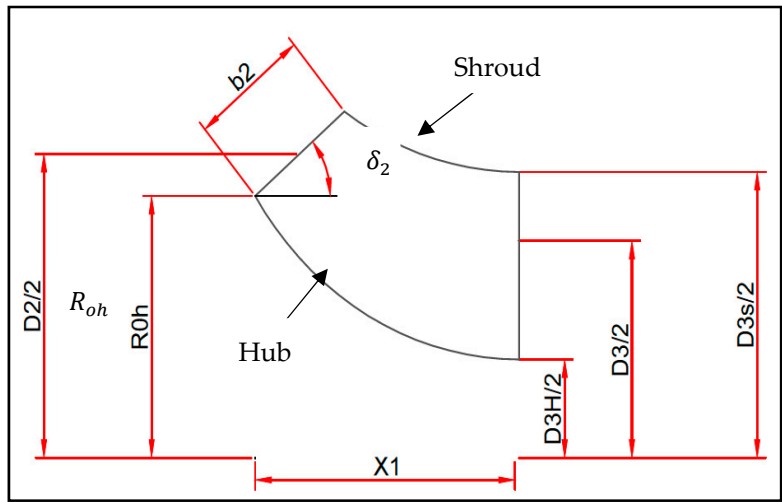

**Figure 1.** The blade Meridian view [42].

The leading edge camber line is obtained by the following relations (Equations (4) and (5)):

$$\theta = \theta_{ref} + \frac{1}{sin(\delta_2)} \int_{x_{ref}}^{x} \tan(\beta_{2b}) \frac{dx}{r} \tag{4}$$

$$r = r_{0h} + (x - x_{0h})\tan(\delta_2) \tag{5}$$

To complete the remainder of the camber line, it is necessary to calculate the coordinates of items 0, 1, *b*, 2, and 3 (Figure 2) given according to the blade dimensions in reference [30], where the Bezier polynomial is used. The following forms are applied, where $x_0$, $x_1$, $x_b$, $x_3$, $\theta_0$, $\theta_1$, $\theta_b$ and $\theta_3$ are axial and radial coordinates of items 0, 1, *b*, and 3 (Equations (6) and (7)).

$$x = (1-u)^4 x_0 + 4u(1-u)^3 x_1 + 6u^2(1-u)^2 x_b + 4u^3(1-u)x_2 + u^4 x_3 \tag{6}$$

$$\theta = (1-u)^4 \theta_0 + 4u(1-u)^3 \theta_1 + 6u^2(1-u)^2 \theta_b + 4u^3(1-u)\theta_2 + u^4 \theta_3 \tag{7}$$

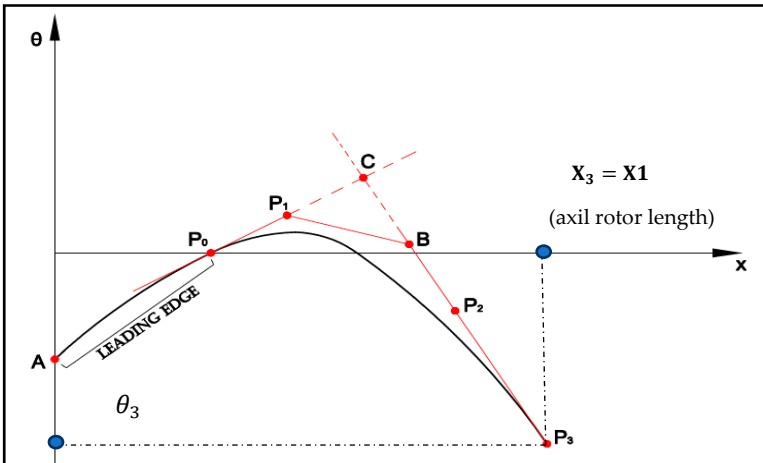

**Figure 2.** The camber line blade view [42].

## 2.2. Grid Discretization and Numerical Method Applied

The CFD object is to solve the fundamental equations that govern flows, the momentum, and the continuity equations. An additional conservation energy equation is solved for flows involving heat transfer phenomena or fluid compressibility. Transport equations are also taken into consideration when the turbulence phenomenon is concerned. The physical problems encountered in this study are described by strongly coupled and non-linear partial differential equations. In general, these equations do not admit analytical solutions except in very simplified cases. This is why recourse to digital resolution methods is necessary. The code used in this study is ANSYS ICEM-CFD, and it is based on the finite volume methods. This numerical resolution method is divisible into two, one phase of mesh and the other is discretization. The mesh phase divides the field of study into small volumes called control volumes. There exist two types of grids, the first is a structured grid, and the second is an unstructured grid. This study used the unstructured hexahedral grid because it adapts well with turbomachine simulations (Figure 3). To reach more precise results, the grid meadows of walls are refined (Figure 4) in object to obtain more information on the thermodynamic parameters in the boundary layer. The ANSYS ICEM CFD code gets the inter-blade channel geometry with its mesh. The discretization phase transforms the continuous problem into a discrete problem: discrete equations and conditions approximate equations and boundary conditions.

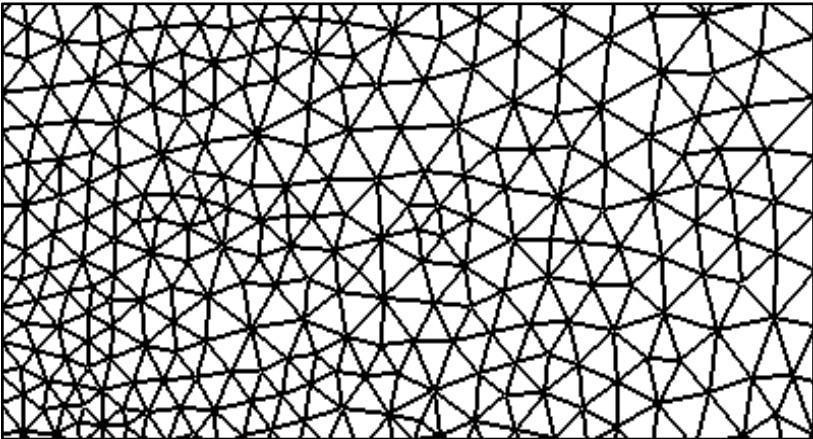

**Figure 3.** The unstructured mesh views in the channel middle.

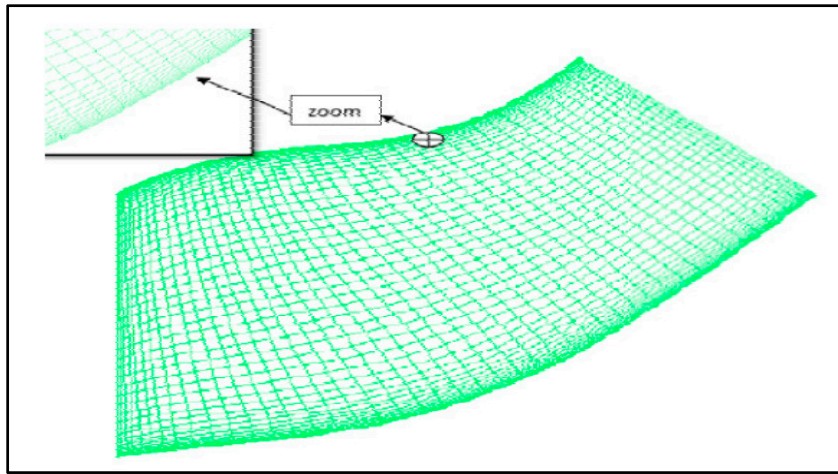

**Figure 4.** The mesh refinement near the wall [42].

*2.3. Flow Simulations in a Mixed Inflow Turbine*

The three-dimensional Navier–Stokes equations consist of conservation equations of mass (8), momentum (9), and energy (10), supplemented by the ideal gas laws (11). The flows in the turbomachines are mainly gaseous and generally consist of air, which allows us to neglect the volume forces in the system of equations.

$$\frac{\partial}{\partial t}\iiint \rho.dv + \iint \rho V.dA = 0 \tag{8}$$

$$\iint (\rho V.dA)V + \iiint \frac{\partial(\rho V)}{\partial t}.dv = \iiint \rho f.dv - \iint p.dA \tag{9}$$

$$\iiint q\rho.dv - \iint pV.dA + \iiint \rho(f.V)dv = \iiint \frac{\partial}{\partial t}\left[\rho\left(e + \frac{V^2}{2}\right)\right]dv + \iint \rho(e + \frac{V^2}{2})V.dA \tag{10}$$

$$p = \rho RT \tag{11}$$

Generally, all the flows in the turbomachines are turbulent. The RANS method is used, and the statistical treatment of turbulence in this method assumes that each instantaneous turbulent quantity can be decomposed into a mean amount and a turbulent fluctuation. This decomposition in the middle and in the fluctuating parts, applied to the Navier–Stokes equations, brings up new terms related to turbulence. The loss of information by the average operation leads to a system with more unknowns than equations and which requires closure models to complete it. The standard *k-ε* turbulence model is used because it gives numerical results closer to experimental results in turbomachinery flows; specifically, that the $y+$ value for the near-wall node has to be in the range of 20 to 100. This model is based on the eddy viscosity concept, which assumes that the Reynolds stresses $-\rho\overline{u_i u_j}$ can be expressed in terms of the mean velocity gradients and the eddy or the turbulent viscosity $\mu_t$, in a manner analogous to the viscous stresses $\tau_{ij}$ for laminar Newtonian flows:

$$\tau_{ij} = \mu\left(\frac{\partial U_j}{\partial X_i} + \frac{\partial u_i}{\partial X_j}\right) - \frac{2}{3}\mu\delta_{ij}\frac{\partial U_k}{\partial X_k}, \tag{12}$$

$$-\rho\overline{u_i u_j} = \mu_t\left(\frac{\partial U_j}{\partial X_i} + \frac{\partial u_i}{\partial X_j}\right) - \frac{2}{3}\mu_t\delta_{ij}\frac{\partial U_k}{\partial X_k} - \frac{2}{3}\delta_{ij}\rho_k \tag{13}$$

This model assumes that the eddy viscosity $\mu_t$ is related to the turbulent kinetic energy $k$ and its dissipation rate $\varepsilon$ presented by the relation (14):

$$\mu_t = \rho C_\mu \frac{k^2}{\varepsilon} \tag{14}$$

where $C_\mu = 0.09$ and $k$, $\varepsilon$ are defined by the Equations (15) and (16):

$$\frac{\partial(\rho k)}{\partial t} + \nabla.\left(\rho k \vec{U}\right) = \nabla.\left[\left(\mu + \frac{\mu_t}{\sigma_k}\right)\nabla k\right] + P_k - \rho_\varepsilon \tag{15}$$

$$\frac{\partial(\rho\varepsilon)}{\partial t} + \nabla.\left(\rho\varepsilon \vec{U}\right) = \nabla.\left[\left(\mu + \frac{\mu_t}{\sigma_k}\right)\nabla\varepsilon\right] + \frac{\varepsilon}{k}(C_{1\varepsilon}P_k - C_{2\varepsilon}\rho_\varepsilon). \tag{16}$$

In this model $P_k$ is the turbulence production and $\sigma_k = 1.00$, $\sigma_\varepsilon = 1.30$, $C_{1\varepsilon} = 1.44$, $C_{2\varepsilon} = 1.92$ are constants determined experimentally.

### 2.4. The Boundary Conditions

The calculation field is an inter-blade channel, the fluid is an ideal gas, with the stationary mode; its speed is equal to 98,000 rpm, and the k-$\varepsilon$ was used as a turbulence model. The inlet mode is subsonic, where the totals pressure and temperature are, respectively, equal to 2.91 bars, 920 K with an absolute flow angle of $-13°$. At the exit, the static pressure is equal to 1 bar, the heat transfer mode of the walls is adiabatic; the blade (intrados and extrados) was considered a dynamic wall; the shroud and the hub as the static wall with the periodicity condition for periodic walls. Figure 5 presents the different parts of the boundary condition.

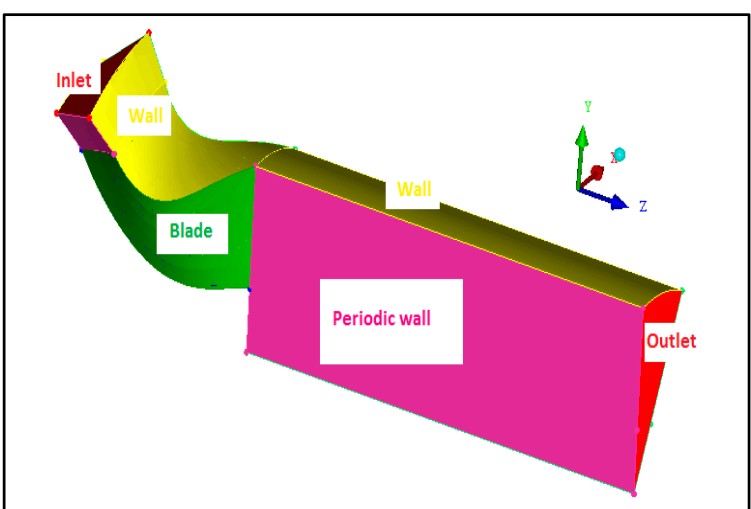

**Figure 5.** The different parts of the boundary condition.

### 2.5. The Mesh Optimization and Grid Generation Solution Dependency

The generation of the mesh (2D or 3D) is a significant phase in a CFD analysis. The influence of its parameters on the calculated solution was given. Excellent mesh quality is essential for obtaining an accurate, a robust, and a meaningful calculation result. The mesh specification depends on the complexity of the geometry and the simulation code used. Four mesh elements were examined (107,244, 233,844, 333,372, and 415,030). The graphical analysis shows that the element number does not influence the torque and the mass flow, a negligible influence on the static pressure distribution (Figures 6a,b and 7).

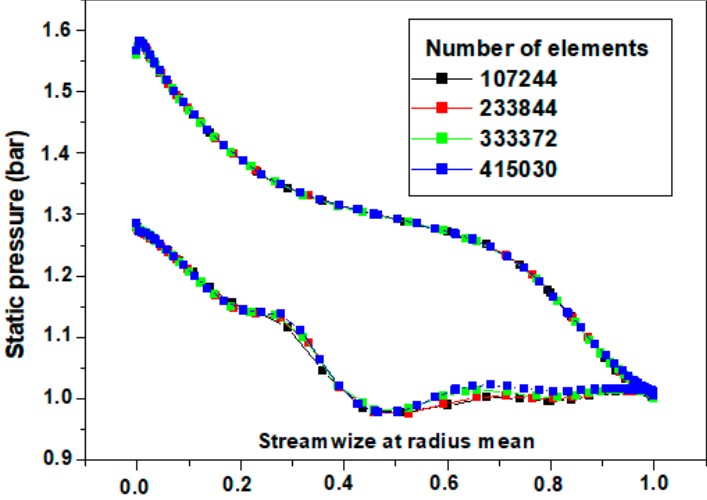

**Figure 6.** Effect of element number on, torque (**a**), mass flow rate (**b**), and efficiency (**c**).

**Figure 7.** Effect of element number on static pressure [42].

On the other hand, a variation in the total to static isentropic efficiency graph stabilizes with element numbers greater than or equal to 333,372 elements used in numerical simulations (Figure 6c). The turbulence model reliability of the mesh number used

(333,372 elements) has been verified by the Y+ contours on intrados and extrados, where the majority of values Y+ are between 20 and 100 (Figure 8).

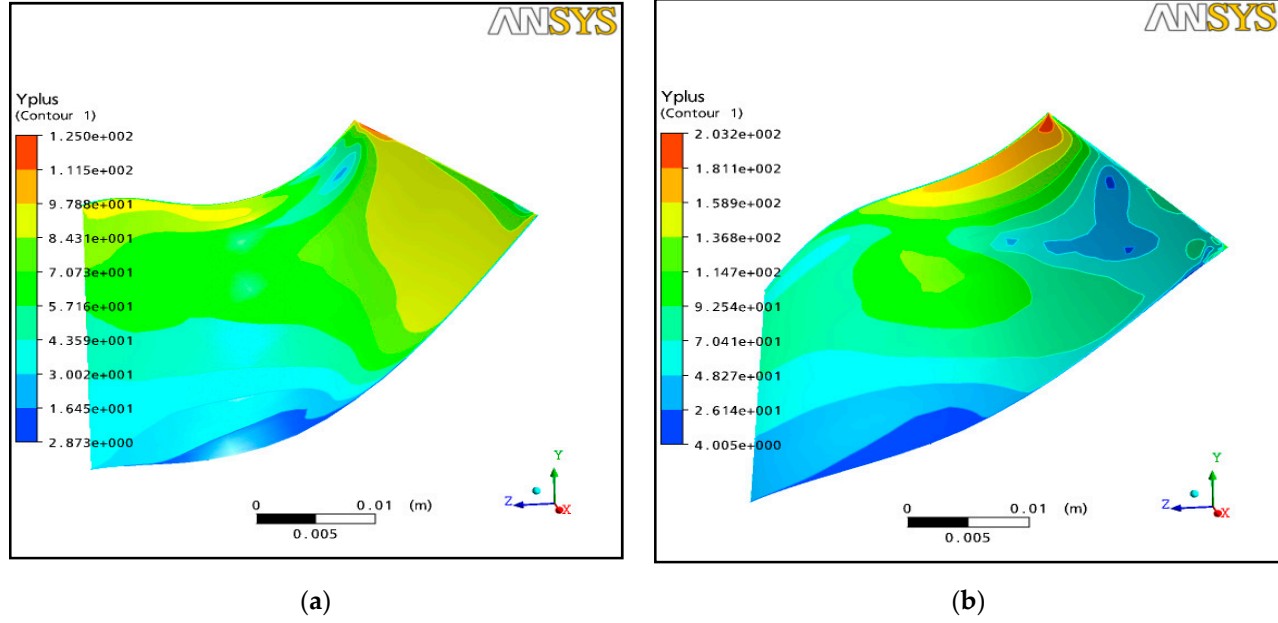

(**a**)                                            (**b**)

**Figure 8.** Y+ contours: (**a**) intrados; (**b**) extrados.

### 2.6. Numerical Model Validations

The numerical validation was confirmed by the experimental works of Chen and Abidat [30] on rotor type A with the geometrical parameter values shown in Table 1. The pressure ratio is defined as the total pressure at the inlet relative to the static pressure at the outlet; the distribution is in the axial direction of the blade. The results are shown in Figure 9. A good reconciliation between the numerical simulation and the experimental results is remarked.

**Table 1.** The geometrical parameter values (mm).

| $b_2$ | $D_2$ | R0h | X1 | D3H | D3 | D3s | $\delta_2(°)$ | $\theta_3(°)$ |
|-------|-------|-----|----|-----|----|-----|------|------|
| 17.99 | 83.58 | 36 | 40 | 27.07 | 59.7 | 78.65 | 40 | −25 |

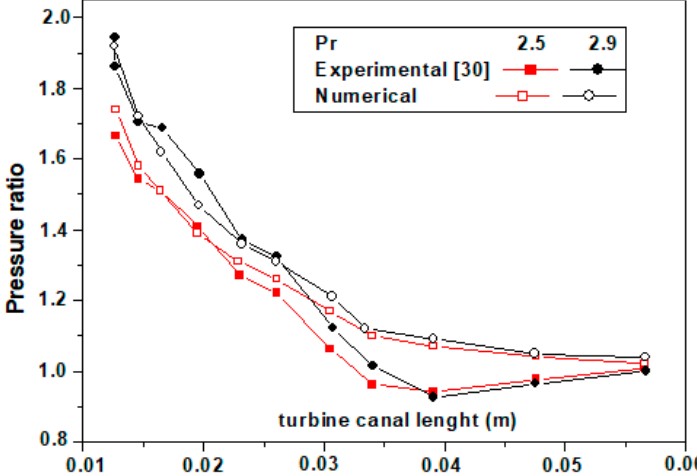

**Figure 9.** Numerical results validation [42].

### 2.7. The Flow Examination of Mixed Inflow Turbine

The static pressure distribution on the blade surface presented in Figure 10 shows that the blade load is higher on the shroud side than on the hub side. Zero blade loads were noticed due to a change in the flow incident near the hub at the leading edge.

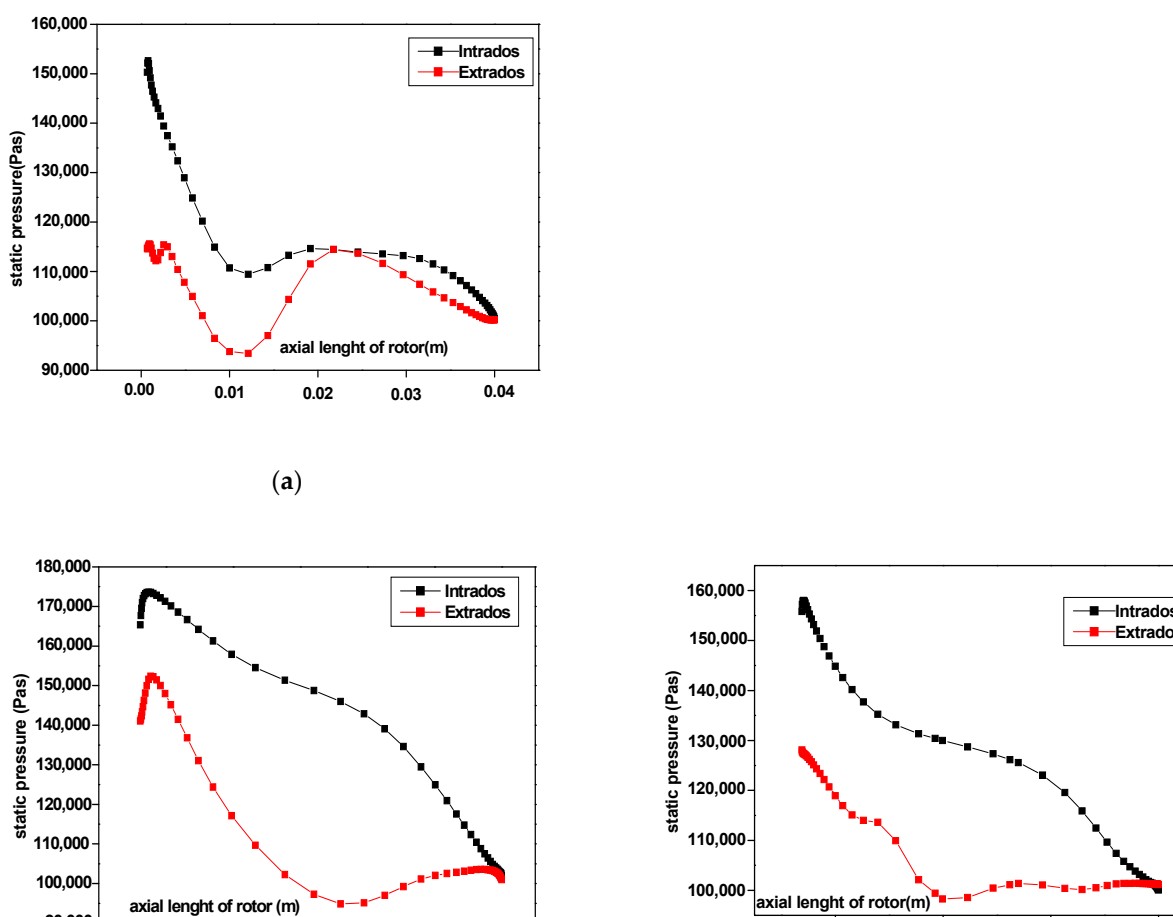

**Figure 10.** The pressure distributions around the blade profiles: (**a**) static pressure near the hub; (**b**) static pressure near the shroud; (**c**) static pressure in the mean radius.

Figure 11 shows that at the intrados, the pressure gradient is almost positive in the radial direction while negative in the axial direction; this proves that the pressure is uniformly distributed in this part of the blade. Contrary to the intrados, the pressure variation on the extrados is really disturbed. This confirms a "pocket" of recirculation within this zone.

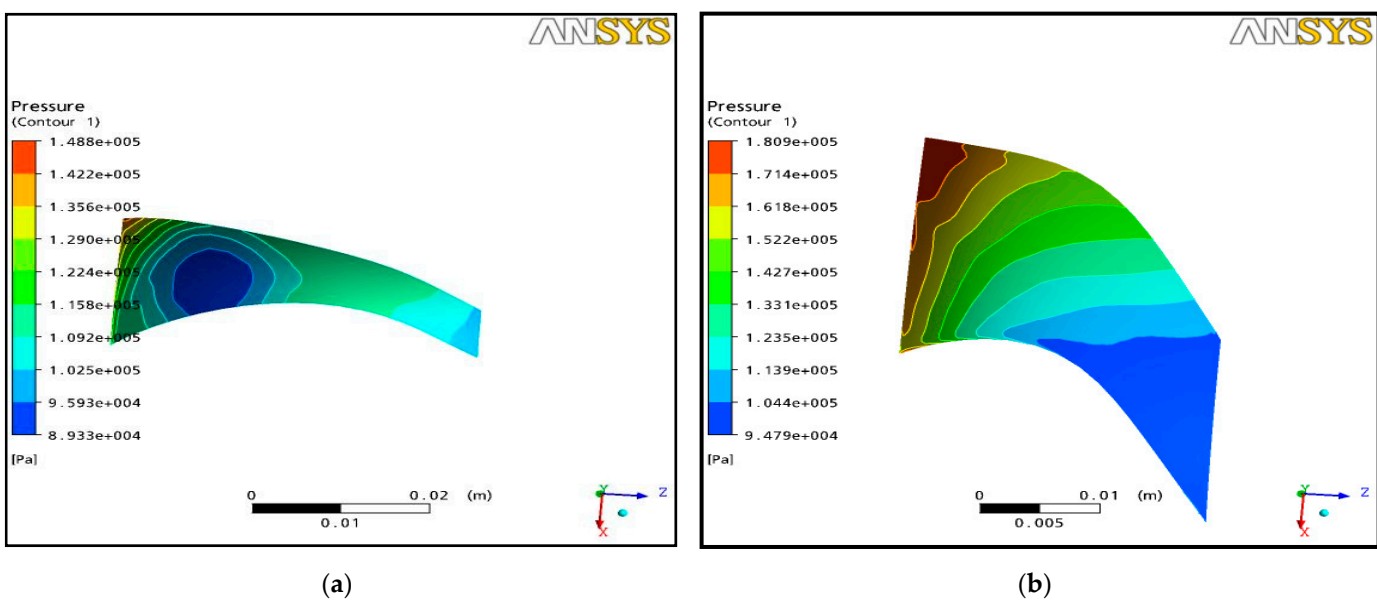

**Figure 11.** Static pressure contours: (**a**) hub; (**b**) shroud.

The vortices density was increased along the rotor length, which reduces the relative tangential speed of the inter-blade flow, as shown in Figure 12. A decrease in the absolute speed along the rotor in the mean meridian plane was remarked in Figure 13 because of the work exchange at the blades level.

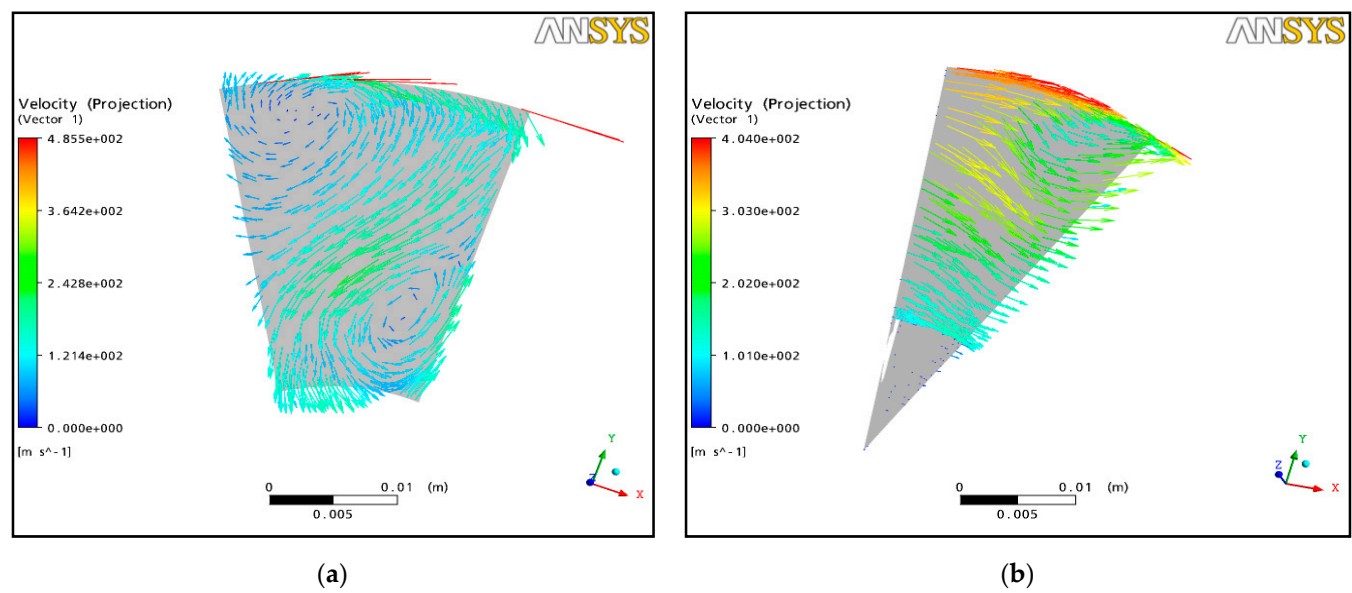

**Figure 12.** Tangential relative velocity field: (**a**) tangential relative velocity field in rotor inlet; (**b**) tangential relative velocity field in rotor outlet.

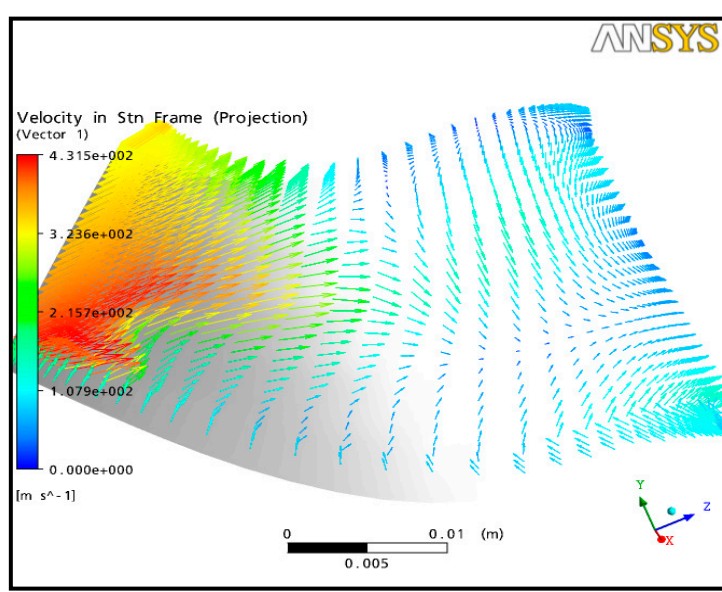

**Figure 13.** The absolute velocity at the mean meridian plane.

### 2.8. The New Design Approach

The new rotor wheels under study are designed by varying the deviation blade angle (−10°, −15°, −20°; −25°; −30° and −35°), which allows for a complete change of the camber line shape as well as the aerodynamic blade shape. The horizontal sliding of the new camber lines revealed that the actual length of the blade increased by increasing the deviation blade angle. Hence, the rotor becomes more prolonged and heavier. The new camber lines blade is shown in Figure 14. The three-dimensional shapes of the blades are configured in Figure 15a,b. The blade characterized by −25° deviation blade angle is considered a reference blade.

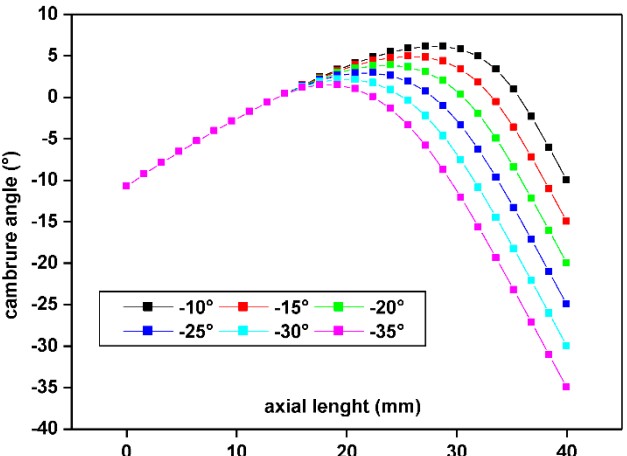

**Figure 14.** The new camber lines blade's view.

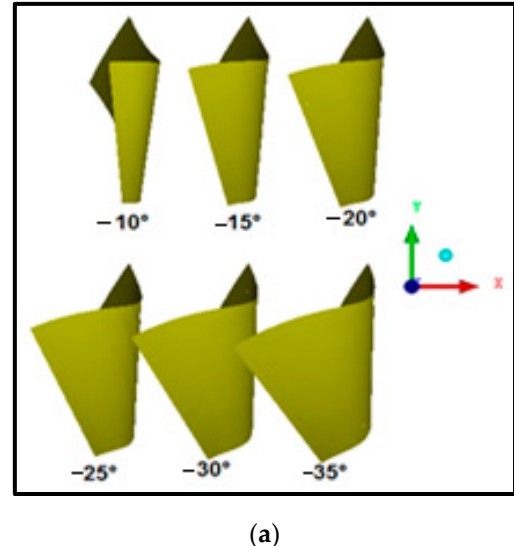
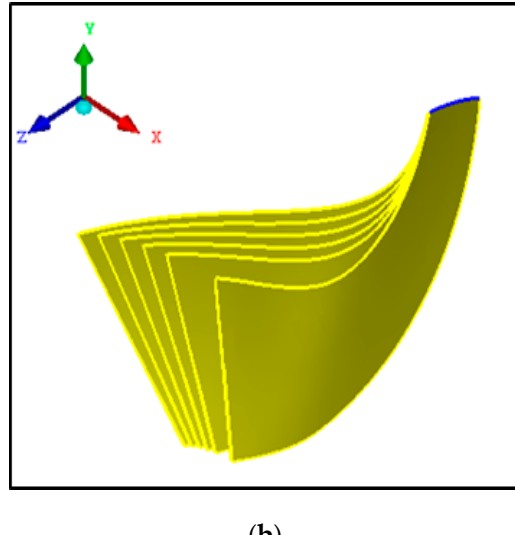

(**a**)                                                  (**b**)

**Figure 15.** The suggested blade design: (**a**) geometrical forms view; (**b**) view of blade geometries comparison.

## 3. Results and Discussions

The investigation of the deviation blade effect is based on the operation of the rotor in three rotation speeds (50%, 75%, and 100% of load), which represent successively 49,000 rpm, 73,500 rpm, and 98,000 rpm. Figure 16a shows that the output work is clearly proportional to the deviation angle for a 100% load and a negligible effect for 50% and 75% loads. The same influence is noted for the case of power (Figure 16b). This behavior is probably due to the rotor surface increase due to the deviation angle (Table 2). Therefore, the exchange surface is larger on one side, and on the other side, the flow adapts with the new blade curvatures.

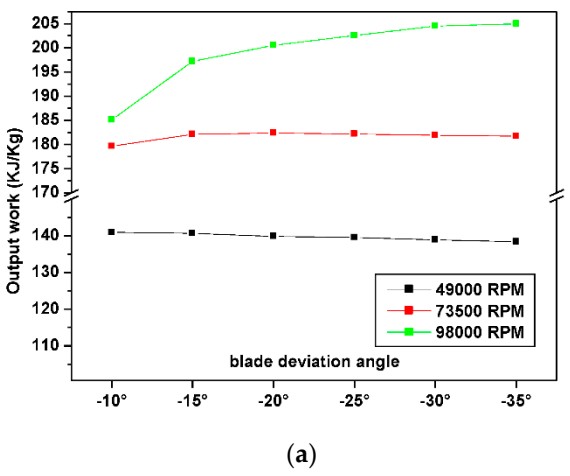
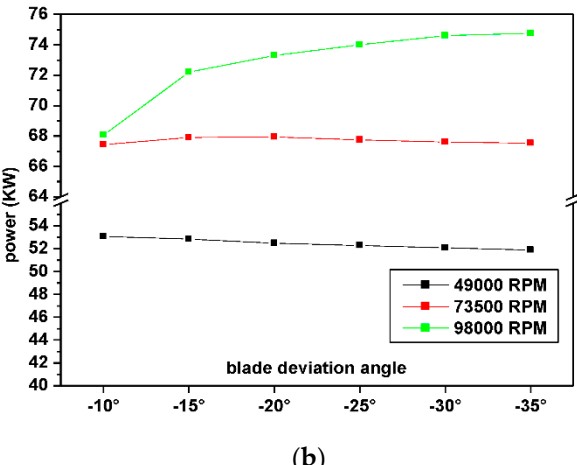

(**a**)                                                  (**b**)

**Figure 16.** The investigation of the deviation blade effect: (**a**) output work for different deviation blade angles; (**b**) power for different deviation blade angles.

**Table 2.** The blade area in mm$^2$ resulted in different deviation blade angle.

| θ3 | −10 | −15 | −20 | −25 | −30 | −35 |
|---|---|---|---|---|---|---|
| A (mm$^2$) | 940 | 970 | 987 | 1006 | 1027 | 1050 |
| Gain A (%) | −6.56 | −3.58 | −1.89 | A-ref | 2.09 | 4.37 |

The analysis of Figure 17, which presents the expansion rate, shows that the deviation blade angle is not clear for a load of 50%; the expansion rate is proportional to the increase in the deviation angle for a 100% load; the opposite is just for a 75% load. The lowest expansion rate corresponds to −10° for a 100% load, and the best expansion rates for value −35 ° are noticed in the case of a 100% load.

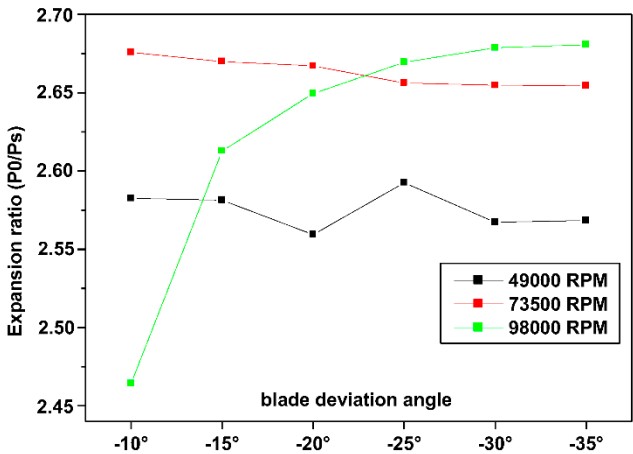

**Figure 17.** Expansion ratio for different deviation blade angles.

Figure 18 shows that the total kinetic energy at the exit rotor is compassionate by the variation of the blade deviation angle; it is essential for the cases of −10°, −15°, and −20° for a rotation speed of 98,000 rpm. A slight influence in the opposite direction for the case of 73,500 rpm and an influence is not evident when 49,000 rpm has been detected.

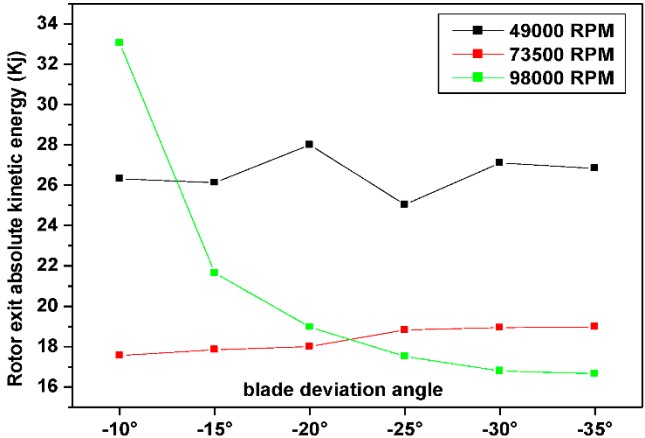

**Figure 18.** Rotor exit total kinetic energy for different deviation blade angles.

The variation of the relative Mach number along the axial direction of the flow for different deviation blade angles at a rotational speed of 50% is presented in Figure 19. It can be noticed that the gas seems to subsonic flow through a diverging–converging channel connection which causes a flow deceleration in the diverging part of the channel to a minimum relative Mach number located at the same axial distance from the entrance, equaling approximately 0.015 mm, followed by flow acceleration in the converging part of the channel. The variation of the relative Mach number between the exit and the rotor entrance is negligible for −10° of deviation blade case, which implies a minimal degree of reaction.

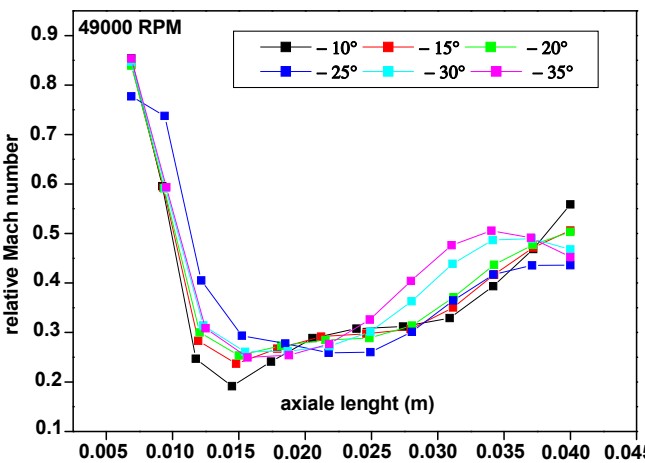

**Figure 19.** Relative Mach number for different deviation blade angles at 50% rotation speed.

The variation is negatively confirmed for the other design cases by falling off the output work magnitude. The variation of the relative Mach number along the axial direction of the flow for different deviation blade angles at a rotational speed of 75% and 100% are successively presented in Figure 20a,b.

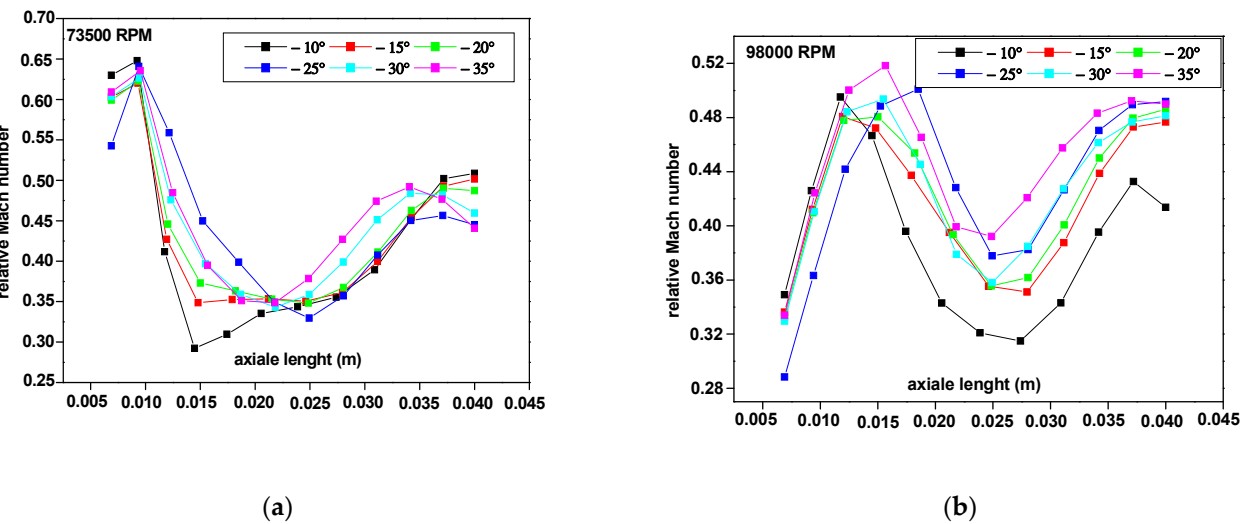

(**a**)                                                                                                                                                         (**b**)

**Figure 20.** The variation of the relative Mach number along the axial direction of the flow for different deviation blade angles: (**a**) relative Mach number for different deviation blade angles at 75% rotation speed; (**b**) relative Mach number for different deviation blade angles at 100% rotation speed.

It can be observed that (Figure 20b) the gas seems to subsonic flow through a converging–diverging channel connection and later through a convergent form which causes acceleration in the converging part of the channel to a maximum relative Mach number—followed by flow deceleration in the diverging part of the channel. All the relative Mach number curves intersect at an axial distance from the inlet within the interval 0.01–0.015 mm, reflecting a starting state of the flow deceleration. Then, there is an acceleration in the convergent blade channel from an axial length of 0.025 m until the exit rotor. The relative Mach number behavior is probably due to the formation of the boundary layer near the blades at several places, making it possible to vary the passage section of the flow, which will cause converging or diverging passages. The pressure distributions around the blade profiles become more uniform with a more significant pressure difference between the pressure side and the suction side, which results in higher loadings at a full rotational speed (Figure 21a–c)).

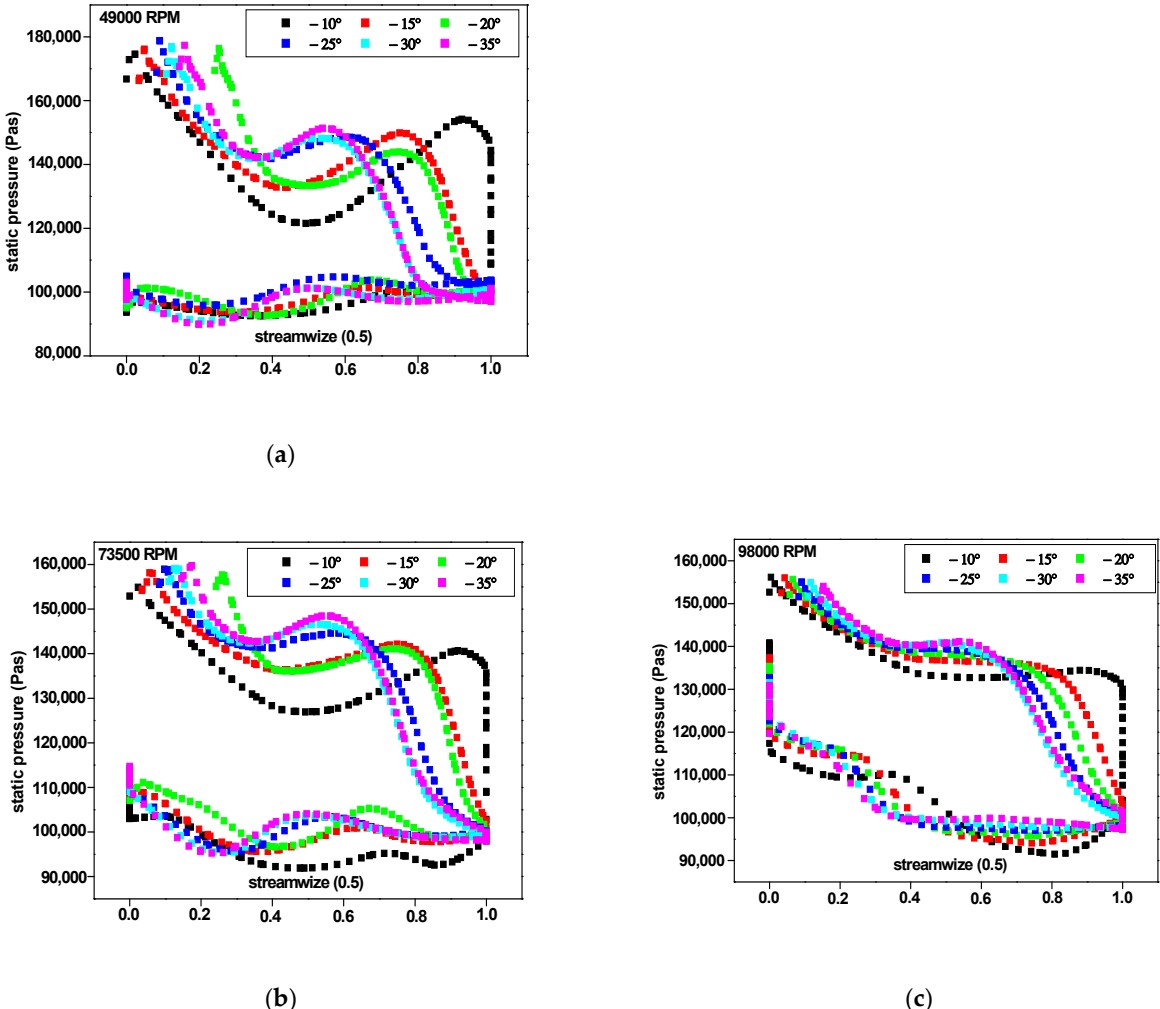

**Figure 21.** The pressure distributions around the blade profiles: (**a**) static pressure for different deviation blade angles in the mean radius at 50% rotation speed; (**b**) static pressure for different deviation blade angles in the mean radius at 75% rotation speed; (**c**) static pressure for different deviation blade angles in the mean radius at 100% rotation speed.

From the total to static isentropic efficiency graph, it is clear that the efficiency is proportional with the increase of the deviation angle for the cases of a 75% and a 100% load but the case of a 50%, which decreased by increasing the deviation angle (Figure 22).

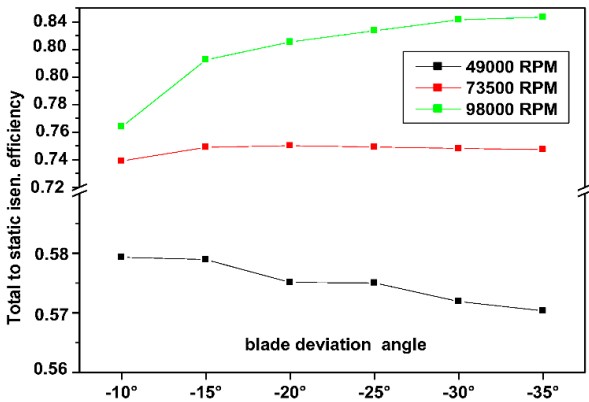

**Figure 22.** Total to static isentropic efficiency for different deviation blade angles.

## 4. Conclusions

A 3D digital model of a mixed-type turbine has been created in this work. Simulation modeling of gas-dynamic and design features of such a turbine has been carried out. By varying the blade deviation angle, some geometric parameters were fixed in order to keep the same rotor casing; several rational geometric shapes of the blade were obtained. It is shown that this approach makes it possible to completely change the three-dimensional shape of the blade and increase the efficiency of the turbine. Geometric tests were carried out at three rotational speeds to demonstrate the effect of deviation blade angle at different machine loads (50%, 75%, and 100%). It has been established that the maximum output work and the resulting power are associated with the highest efficiency for a specific optimal design case of a 100% machine load (deviation blade angle $-35°$), and they are achieved by increasing the contact surface of the blade with the fluid. It was also remarked a negligible effect of blade deviation angle on power, output work and efficiency for 50% and 75% loads. It was observed that the deviation blade angle is not clear for a load of 50%; the expansion rate is proportional to the increase in the deviation angle for a 100% load; the opposite is just for a 75% load. It has been observed that the static pressure distribution will be more uniform in the case of a 100% load. The variation of the relative Mach number along the axial direction of the flow for different deviation blade angles at different rotational speeds showed that the fluid behaviors in the rotor are the same in the converging–diverging nozzle due to the formation of the boundary layer near the blades at several places.

It should be noted that with such a rational turbine geometry, the weight of the rotor increases by only 4.37%, which is quite acceptable. For higher absolute output kinetic energies at deviation blade angles from $-10°$ to $-20°$, it is recommended to use an exhaust diffuser to recover part of it to a larger expansion ratio. The effect of blade deviation angle on mixed flow performance drops sharply starting at $-20°$ at a 100% machine load, so it is recommended to avoid using angles greater than $-20°$ as vane angle values.

The developed digital model can also be effectively used at all subsequent stages of the design and manufacture of turbines, including for use in CAD to create 3D models and drawings, in automated control systems to create a processing technology using CNC equipment, as well as to determine the power and the frequency characteristics in other sections of Ansys. It will significantly reduce the time and the cost of creating a turbine.

**Author Contributions:** Conceptualization, M.A.C. and M.S.; methodology, M.A.C.; software, M.A.C., I.K., I.P. and Y.B.; validation, S.D., I.P. and V.I.; formal analysis, S.D. and M.S.; investigation, M.A.C. and Y.B.; resources, Y.B., V.I. and M.A.C.; data curation, V.I. and I.K.; writing—original draft preparation, M.A.C. and Y.B.; writing—review and editing, S.D., V.I. and I.P.; visualization, M.A.C. and Y.B.; supervision, V.I. and I.K.; project administration, M.A.C., V.I. and I.K.; funding acquisition, I.K. and M.S. All authors have read and agreed to the published version of the manuscript.

**Funding:** This work has been supported by grant agency VEGA project No. 1/0073/19 and grant agency KEGA project No. 001ŽU-4/2020.

**Institutional Review Board Statement:** Not applicable.

**Informed Consent Statement:** Not applicable.

**Data Availability Statement:** Not applicable.

**Acknowledgments:** The general approach has been partially developed within the research project "Development of a methodology for optimal design and manufacture highly efficient, highly reliable turbomachines, taking into account various operating modes" (State reg. no. 0121U107511). The results have been partially obtained within the project "Fulfillment of tasks of the perspective plan of development of a scientific direction "Technical sciences" Sumy State University" funded by the Ministry of Education and Science of Ukraine (State reg. no. 0121U112684). The research was partially supported by the Research and Educational Center for Industrial Engineering (Sumy State University) and International Association for Technological Development and Innovations.

**Conflicts of Interest:** The authors declare no conflict of interest.

## Nomenclature

| | |
|---|---|
| $\vec{B}$ | body Forces |
| $b_2$ | Blade height at rotor inlet |
| $D_2$ | Mean diameter at rotor inlet |
| $D_3$ | Exducer hub diameter |
| $D_{3S}$ | Exducer tip diameter |
| $e$ | Specific internal energy |
| $\vec{f}$ | Friction forces |
| $P$ | pressure |
| $q$ | Heat transfer energy |
| $R$ | Perfect gas constant |
| $r$ | Radial polar variable |
| $R_{0h}$ | The radius at the tip rotor inlet. |
| $t$ | time |
| $T$ | Temperature |
| $U$ | variable between o and 1 |
| $\vec{V}$ | Absolute velocity |
| $w_e$ | The specific external force work |
| $X$ | Axial polar variable |
| $X1$ | Length of the rotor |
| $x_{0h}$ | The axial distance for the initial point of the hub. |
| $x_{ref}$ | Reference axial distance of the blade |
| $\phi$ | The inlet blade angle in the axial and tangent plan |
| $\lambda$ | The flow cone angle |
| $\eta$ | efficiency |
| $\rho$ | Density |
| $\delta_2$ | Cone angle at rotor inlet |
| $\beta_{2b}$ | Mean blade angle at rotor inlet |
| $\theta$ | Camber angle |
| $\theta_3$ | Deviation blade angle |
| $\theta_{ref}$ | Reference camber angle |
| $\nabla P$ | Pressure forces |

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
