# Peer review of "The Effect of Blade Angle Deviation on Mixed Inflow Turbine Performances"

_applsci, doi:10.3390/app12083781_

Round 1

Reviewer 1 Report

General comments: 

The authors compared different blade deviation angles on the geometric blade shape and the performance of semi-axial turbines by ANSYS ICEM CFD. These factors are essential to the blade design. However, overall, the figures in the content are vague. English is also poor. For example, no structured grid should be an unstructured grid. The organization of the manuscript and presentation of the data and results must be improved. Some parts are not even clearly demonstrated. The ?−? model is chosen to simulate the turbulence. In line 175, it is said that y+ for the near-wall node has to be in the range of 20 to 100. What’s your y+ value for three different meshes, respectively? Besides, what are your boundary conditions? I recommend major revision before its acceptance for publishing on Appl. Sci. 

 Specific comments follow with line numbers in brackets:

Figure 1: 0h in the R0h should be in subscript format. Other notation should be the same as Symbols & Description in the end.

Figures: The number color is red, black is recommended.

Line 132: There is no description of x0, x1, xb,x3. x1 is the length of the rotor?

Line 154: There is no description of the specific boundary conditions.

Figure 3: It is difficult to see the mesh is unstructured in the figure.

Equations: dot point in the equations are not in the middle in the vertical direction.

Line 199: From Figure 5, it is noted that the results are close between 107244 and 233844. However, there is little distinction when the mesh number is higher than 233844. Are you sure to use results got from 333372?

Figure 5,6,7: The title includes [42]. Is it extra?

Line 206: First, the mesh optimization is done. Then, numerical model validations are done. However, what is the calculation condition of the mesh optimization?

Figure 13, 14; mach should be Mach in the y-axis.

Line 326: Symbols instead of Symbols.

Author Response

Dear Reviewer,
The authors appreciate your time devoted to reviewing the manuscript and your valuable comments. The answers are prepared and attached in a separate file.

Reviewer 2 Report

The article is a good subject but written at a low level.
So I recommend a reject of the article.

The title of the article completely corresponds to it.
The article simulates the change of deviation blade angles of the turbine.
The main comments to the article:
  • The main remark is that 1/3 of the article is a complete copy of the methodology of the article [42]. However, the authors use different terminology in the results than the one presented in the methodology. For example, "deviation blade angles" - in Article [42] is presented as "Cone angle". And it is also mentioned in the notation. For this article, the authors must completely rework the methodology.
  • All drawings and graphics are of poor quality.
  • There is a comma separation and many small errors in the text.
  • The article has almost no scientific novelty.
  • The simulation results have not been confirmed by experiments.

Author Response

(The authors gave the same response as above.)

Reviewer 3 Report

The authors must justify the choice of the selected turbulence model. They must also justify the values of y+, since for the values they indicate, other turbulence models may be more appropriate.

Author Response

(The authors gave the same response as above.)

Reviewer 4 Report

Comments on the manuscript entitled "Effect of Deviation Blade Angle on Mixed Inflow Turbine Performances" by Chelabi et al. submitted to Applied Sciences

In this manuscript, the authors numerically investigated the effect of deviation blade angle on the performance of the mixed inflow turbine using CFX and obtained the angle for the optimal performance. Comments are as follows: 

  1. Before examining the effect of deviation blade angle on the turbine performance, a case showing the accuracy of the code and setup for predicting the quantities of interest is needed, probably by simulating an existing turbine design.  
  2. The motivation for this study is not properly stated in the introduction. 
  3. The introduction should be structured in a better way instead of a list of papers in the literature with descriptions.
  4. The usefulness of this work for actual turbine design should be discussed in the conclusions section. 
  5. The abstract should be expanded to include the key results from this work. 
  6. Besides the curves, the flow field should be analyzed for people to understand better the mechanism affecting the performance of the turbine. 

Author Response

(The authors gave the same response as above.)

Round 2

Reviewer 1 Report

Thank you for the revision. The revised one directly deal with my concerns. The quality of the figures are improved. However, there are still space to be improved for this manuscript, such as the sequence in Symlols & Description.

Best regards,

Wei

Author Response

Dear Reviewer,
The authors appreciate your comments and updated the paper.

Reviewer 2 Report

So far, the authors have not improved the article much, so I recommend major revision.

Comments:
- All the figures remained of poor quality.
- In most cases, the comma-separated numbers in all figures remain (5,6,8,9,14,16,17,18,19).
- The description of the results obtained and the schedule are very limited and need further improvement (eg subsection 2.6).
- There are many problems with the design and structure of the article.

Author Response

(The authors gave the same response as above.)

Reviewer 4 Report

This reviewer does not think the authors have seriously taken into account the comments on the first version of the manuscript. This reviewer cannot suggest publication of this work in the present work. 

  1. For the motivation of the paper, what the authors added is more like an objective.
  2. An introduction simply listing all the relevant literature cannot help the readers to understand the status of the field, and the background and the motivation of this work. 
  3. The discussion on the usefulness of this work is shallow. The discussion should be beyond the result itself and be related to the real-life design.
  4. Besides the pressure on the surface, the velocity filed is suggested to examine. 

Author Response

(The authors gave the same response as above.)

Round 3

Reviewer 4 Report

  1. The long paragraph starting by "The geometric shape complicity of the blade has limited ..." must be structured in a better way.
  2. The considered scenario is relatively simple. The limitations of present work should be discussed in the conclusions section. 

Author Response

Dear Reviewer,

The authors appreciate your comments and time devoted to reviewing this manuscript. The answers to your comments are attached in a separate file.

Best regards,
Authors
